# *MCP-1* rs1024611 Polymorphism, MCP-1 Concentrations, and Premature Coronary Artery Disease: Results of the Genetics of Atherosclerotic Disease (GEA) Mexican Study

**DOI:** 10.3390/biomedicines12061292

**Published:** 2024-06-11

**Authors:** Rosalinda Posadas-Sánchez, Fernando Velázquez-Sánchez, Juan Reyes-Barrera, Guillermo Cardoso-Saldaña, Frida Velázquez-Argueta, Neftali Eduardo Antonio-Villa, José Manuel Fragoso, Gilberto Vargas-Alarcón

**Affiliations:** 1Department of Endocrinology, Instituto Nacional de Cardiología Ignacio Chávez, Mexico City 14080, Mexico; rossy_posadas_s@yahoo.it (R.P.-S.); med.fvs@gmail.com (F.V.-S.); reyesbarrera_juan@hotmail.com (J.R.-B.); gccardosos@yahoo.com (G.C.-S.); frida.var.b@gmail.com (F.V.-A.); neftalivilla@comunidad.unam.mx (N.E.A.-V.); 2Department of Molecular Biology, Instituto Nacional de Cardiología Ignacio Chávez, Mexico City 14080, Mexico; mfragoso1275@yahoo.com.mx; 3Research Direction, Instituto Nacional de Cardiología Ignacio Chávez, Mexico City 14080, Mexico

**Keywords:** inflammation, genetic susceptibility, monocyte chemoattractant protein-1, polymorphisms, premature coronary artery disease

## Abstract

Monocyte chemoattractant protein-1 (MCP-1) participates in the initiation and progression of atherosclerosis. In vitro studies have reported that the *MCP-1* rs1024611 polymorphism is associated with increased MCP-1 concentrations. The study aimed to define whether MCP-1 concentrations are associated with premature coronary artery disease (pCAD) and to establish whether variations in the rs1024611 polymorphism increase MCP-1 concentrations. *MCP-1* rs1024611 polymorphism was determined in 972 pCAD patients and 1070 control individuals by real-time PCR. MCP-1 concentrations were determined by the Bio-Plex system. In the total population, men had higher MCP-1 concentrations when compared to women (*p* < 0.001). When stratified by rs1024611 genotypes, higher MCP-1 concentrations were observed in *AA* individuals compared to *GG* subjects (*p* = 0.023). When performing the analysis considering sex, the differences remained significant in women (*AA* vs. *GG*, *p* = 0.028 and *GA* vs. *GG*, *p* = 0.008). MCP-1 concentrations were similar in pCAD patients and controls (*p* = 0.782). However, the independent analysis of the studied groups showed that in patients with the *AA* genotype, MCP-1 concentrations were significantly higher when compared to patients with the *GG* genotype (*p* = 0.009). Considering that the *AA* genotype increases MCP-1 concentration, we evaluated whether, in *AA* genotype carriers, MCP-1 concentrations were associated with pCAD. The results showed that for every ten pg/mL increase in MCP-1 concentration, the risk of presenting pCAD increases by 2.7% in *AA* genotype individuals. Individuals with the *MCP-1* rs1024611 *AA* genotype present an increase in MCP-1 concentration. In those individuals, increased MCP-1 concentrations increase the risk of presented pCAD.

## 1. Introduction

Coronary artery disease (CAD), a clinical manifestation of atherosclerosis, is considered one of the principal causes of morbidity and mortality worldwide [1]. The role of inflammation in the progression of atherosclerosis is well known. Infiltration of several cell types, including T cells and macrophages, has been reported in atherosclerotic plaque [2]. The production of cytokines, chemokines, and growth factors by these cells perpetuates the damage in the atherosclerotic lesion [3]. Monocyte chemoattractant protein (MCP)-1/CCL2 is a chemokine with an important role in the formation, progression, and destabilization of atherosclerotic plaque. After an infarction, this chemokine participates in tissue remodeling. MCP-1 concentrations increase in peripheral blood after myocardial infarction, both in animal models [4] and in humans [5,6,7], and correlate with the prognosis of the disease [8]. It is well-known that environmental and genetic factors, as well as interactions between them, participate in triggering this pathology [9,10]. Studies in families and twins have indicated that the heritability of CAD varies between 40 and 60% [11]. This heritability has been seen most importantly in early-onset or pCAD [12]. The study of genetic participation in the development of CAD has focused on the variations at the DNA level, mainly single nucleotide variations (SNP). Genome-wide studies (GWAS) and candidate genes have established the participation of various SNPs in the development of CAD [13,14,15]. Unfortunately, GWAS studies do not include Mexican populations and candidate gene studies in Mexicans are scarce [16,17].

In 1999, a biallelic *A*/*G* polymorphism was described in the distal regulatory region at position -2518 (rs1024611) of the *MCP-1* gene. This polymorphism affects the expression level of MCP-1 in response to the inflammatory stimulus [18]. Furthermore, previous data suggest that this polymorphism is involved in the pathogenesis of CAD [19]. However, the data in CAD are controversial, with positive and negative results [20,21]. The problem with these studies is the inclusion of a small group of individuals, both patients and controls, as well as the selection of control individuals, which has been carried out based on clinical questionnaires. Thus, the objective of the present study is to define if MCP-1 concentrations and MCP-1 rs1024611 genotypes are associated with the presence of pCAD in Mexican mestizo individuals from the GEA (Genetics of Atherosclerotic Disease) Study.

## 2. Materials and Methods

### 2.1. Subjects

The study included 972 patients with pCAD and 1070 healthy controls, all belonging to the GEA Mexican Study (Figure 1). pCAD was defined when the individual presented with acute myocardial infarction, revascularization surgery, angioplasty, or coronary stenosis > 50% on angiography and when the diagnostic was made before the of age 55 years in men and before the age of 65 years in women. The control group was selected from subjects who went to the Institute’s blood bank for donation purposes, or from those who were invited to participate through posters or by direct invitation in social centers. This group included healthy individuals with no family or personal history of pCAD. A computed tomography scan of the chest and abdomen was performed on all participants. Coronary artery calcification (CAC) was defined using the Agatston method [22]. All individuals included in the control group had a CAC score equal to zero. Demographic, clinical, and biochemical parameters and lifestyle characteristics were assessed in all participants and defined as previously described [23,24,25].

To evaluate the influence of population stratification, 265 informative ancestry markers that distinguish Amerindian, European, and African ancestry were determined in all individuals [16]. Similar overall ancestry was observed in study individuals with 54.0% Amerindian, 35.8% Caucasian, and 10.1% African ancestry in controls, and 55.8% Amerindian, 34.3% Caucasian, and 9.8% African ancestry in patients with pCAD (*p* > 0.05).

### 2.2. MCP-1 Concentrations

A Bioplex system (R&D Systems, Minneapolis, MN, USA) was used to determine the serum MCP-1 concentrations. Concentrations were determined in thawed samples obtained between 2008 and 2013. The data were analyzed with the Bio-Plex Manager software (version 4.3). Results are expressed in pg/mL.

### 2.3. MCP-1 rs1024611 Genotypes Determination

Genomic DNA was extracted from whole blood (containing EDTA) using the QIAamp DNA Blood Mini kit (QIAGEN, Hilden, Germany). The genotypes of the *MCP-1* rs1024611 polymorphism were determined using TaqMan assays on a model 7900HT real-time PCR machine (Applied Biosystems, Foster City, CA, USA). To ensure that the determinations were correct, 10% of the total samples were determined in duplicate, and the results were 100% concordant.

### 2.4. In Silico Analysis

To determine the possible functional effect of the rs1024611 polymorphism of the *MCP-1* gene, we used the GWASTools bioconductor program (https://www.bioconductor.org/packages/release/bioc/html/GWASTools.html) (accessed on 5 September 2023). This program predicts whether the alleles of the polymorphisms create or destroy binding sites for transcription factors, splicing-regulating proteins, and microRNAs, or whether the alleles affect the structure and function of the MCP-1 protein.

### 2.5. Statistical Analysis

Before performing the descriptive analysis, we test the distribution of the continuous variables with the Kolmogorov–Smirnov test. When the *p*-value was <0.05, the variable distribution was considered asymmetrical. Data are expressed as frequencies or median (interquartile range). The Mann–Whitney U test was used for comparisons of continuous variables, while the Chi-square test was used for comparisons of categorical variables. Allele and genotype frequencies were determined by direct counting. It was determined whether the rs1024611 polymorphism was in the Hardy–Weinberg equilibrium using the Chi-square test. The association of MCP-1 concentrations with pCAD in individuals with the *AA* genotype of the rs1024611 polymorphism was evaluated by logistic regression analysis adjusting for potentially confounding variables. The analysis of MCP-1 concentrations in the groups studied and stratified by sex or by the genotypes of the rs1024611 polymorphism was performed using the Mann–Whitney U or Kruskal–Wallis tests. We considered significant differences when the *p*-value was <0.05.

## 3. Results

### 3.1. Demographic, Clinical, and Biochemical Characteristics

There are demographic, clinical, and biochemical differences in the study groups. The patients are older, are preferably men, and have a higher body mass index (BMI). Other differences between the groups are shown in Table 1.

### 3.2. MCP-1 Concentrations in the Study Groups

MCP-1 concentrations were similar in patients with pCAD [216 pg/mL (157–293)] and control individuals [218 pg/mL (169–289)] (*p* > 0.05). However, in the total group, men [225 pg/mL (165–300)] had higher concentrations of MCP-1 than women [208 pg/mL (151–284)] (*p* < 0.001) (Figure 2A). When stratifying MCP-1 concentrations by the genotypes of the rs1024611 polymorphism, individuals with the *AA* genotype [224 pg/mL (164–300)] presented higher concentrations than carriers of the *GG* genotype [210 pg/mL (156–283)] (*p* = 0.023) (Figure 2B). This difference remained significant when making the comparison considering the dominant model (*GG* vs. *GA* + *AA*) (*p* = 0.029) (Figure 2C). The distribution of the genotypes of the studied polymorphism was in the Hardy–Weinberg equilibrium.

When performing the analysis separately in women and men, the differences remained significant in women. Carriers of the *AA* [215 pg/mL (156–285)] and *GA* [216 pg/mL (154–292) genotypes have higher MCP-1 concentrations when compared to carriers of the *GG* [190 pg/mL] genotype (*p* = 0.028 and *p* = 0.008, respectively) (Figure 3A). This difference remained significant under the dominant model (*p* = 0.004) (Figure 3B).

When analyzing MCP-1 concentrations independently in patients with pCAD and controls, in patients the concentrations were different in the genotypes (*p* = 0.028). It was observed that patients with the *AA* genotype presented higher concentrations of MCP-1 [234 pg/mL (158–322)] when compared to carriers of the *GG* genotype [209 pg/mL (155–284)] (*p* = 0.009) (Figure 4A). In this group, the differences remained significant in the group of women. Carriers of the *AA* genotype [257 pg/mL (147–298)] had the highest concentrations of MCP-1 when compared to carriers of the *GG* genotype [172 pg/mL (129–272)] (*p* = 0.039) (Figure 4B). The difference remained significant when doing the analysis considering the dominant model (*p* = 0.032) (Figure 4C).

### 3.3. Association of MCP-1 Concentrations with pCAD in Individuals with the rs1024611 AA Genotype

Considering that the rs1024611 *AA* genotype causes high concentrations of MCP-1, we evaluated whether, in carriers of this genotype, MCP-1 concentrations were associated with the presence of pCAD. The logistic regression analysis was adjusted by different models. Model 1 by age and sex, model 2 by age, sex, BMI, and smoking, model 3 by age, sex, BMI, smoking, LDL-cholesterol (LDL-C), and pattern B of LDL and model 4 by age, sex, BMI, smoking, LDL-C, LDL pattern B, hypertension and inflammation (C-reactive protein values > 3 mg/L) (Figure 5). In the model with the greatest adjustment (model 4), for every 10 pg/mL increase in MCP-1 concentration, the risk of presenting pCAD increases by 2.7% in individuals with the *AA* genotype.

## 4. Discussion

MCP-1 is a monocyte chemoattractant protein that participates in the formation, progression, and destabilization of atheroma plaques in the atherosclerotic process. In a Mexican cohort with a significant number of individuals well characterized from the anthropometric, biochemical, clinical, demographic and tomographic point of view, we reported high concentrations of this chemokine in individuals carrying the *AA* genotype of the *MCP-1* rs1024611 polymorphism. The increase remained significant in the group of women. MCP-1 concentrations were similar in patients with pCAD and controls. However, the independent analysis of the groups studied showed that in patients with the *AA* genotype, MCP-1 concentrations were significantly higher when compared to patients carrying the *GA* and *GG* genotypes. We report that, for every 10 pg/mL increase in MCP-1 concentration, the risk of presenting pCAD increases by 2.7% in individuals with the *AA* genotype.

Various studies have established the role of inflammation in the development of CAD (2,3). Inflammation is relevant in various stages of the disease, from the beginning to the progression of atheroma, being associated with plaque instability and rupture [26]. Several chemokines participate in the recruitment of leukocytes (monocytes, macrophages, and T lymphocytes) in the process of atherosclerotic plaque formation [27]. Monocytes/macrophages play a relevant role in the initiation, progression, and complications of coronary atherosclerosis [28,29]. MCP-1 is mainly responsible for the recruitment of monocytes to active sites of inflammation and, therefore, in the development of atheroma [30]. Studies that have analyzed MCP-1 concentrations in patients with CAD have shown contradictory results. On the one hand, higher MCP-1 concentrations have been reported in patients with CAD than in control groups, and in other cases, concentrations have been similar in patients and controls [31,32,33]. In our study group, patients with pCAD and control individuals presented similar concentrations of MCP-1. The differences reported in the studies may be due to the participation of various polymorphisms in the gene that encodes this chemokine and that could be conditioning its concentrations. Given this, we analyzed a polymorphism, rs1024611, that has previously been associated with variations in MCP-1 concentrations [18,34]. When analyzing the total group of individuals (patients and controls), we detected that individuals with the *AA* genotype present higher concentrations of MCP-1 compared to carriers of the *GG* genotype. This difference remained significant in the women group. When the patient and control groups were analyzed independently, patients with the *AA* genotype presented higher concentrations of MCP-1 when compared to patients carrying the *GG* genotype. Thus, both in the total group and in the patients, carriers of the *AA* genotype presented higher concentrations of MCP-1 when compared to carriers of the *GG* genotype. The rs1024611 polymorphism is located in the regulatory region of the gene and according to the bioinformatics analysis we performed, the variation in this polymorphism modifies the binding affinity of the transcriptional factors JUN, AP-4, IRF1, and CEBP. In this case, the presence of the *G* allele in this position decreases the affinity for these transcriptional factors, leading to a decrease in gene expression and consequently generating fewer concentrations of MCP-1. In our study, individuals with the *AA* genotype presented higher concentrations of MCP-1 than those carrying the *GG* genotype, this being in accordance with the functional prediction that we made for this polymorphism. Studies by Fenoglio et al., Gonzalez et al., Rovin et al., McDermott et al., Tabara et al. [18,34,35,36,37] have reported high levels of MCP-1 in individual carriers of the *GG* genotype. Some differences can be observed in these studies compared with our work. First, Rovin’s work was an in vitro study using the luciferase assay and peripheral blood mononuclear cell cultures [18]. In both cases, the assays used IL-1beta to induce the production of MCP-1. On the other hand, Fenoglio et al., and McDermott et al. [35,37] studied Caucasian individuals, Tabara et al. [34] studied Japanese individuals, and Gonzales et al. [36] analyzed an admixture population (European-, African-, and Hispanic-Americans); however, only 6% were Hispanic-Americans. The number of individuals analyzed with MCP-1 levels and rs1024611 genotypes vary, the study by Gonzalez et al. [36] included 36 individuals with rs1024611 *AA* genotype and 41 with *AG + GG* genotypes, the study by McDermott et al. [37] included 122 with *GG* and 1480 with *GA + AA*, the Tabara et al. [34] study included 47 with *AA* and 278 with *GA + GG*, and the Feroglio et al. [35] study included 122 individuals with no information about the number of individuals with different genotypes. Contrary to those results, a recent study reported that COVID-19 patients with the rs1024611 *AA* genotype presented higher concentrations of MCP-1 than a control group; however, they did not detect differences with the *GG* and *AG* genotypes [38]. The same was observed by Hwang et al. [39], who compared the amount of MCP-1 in the sera of patients with *GG*, *GA*, and *AA* genotypes and found no significant differences among the groups. It is important to notice that the frequency of the *MCP-1* rs1024611 *A* and *G* alleles varies in different populations. According to data obtained from the National Center for Biotechnology Information (https://www.ensembl.org/index.html accessed on 9 February 2024), the frequency of the rs1024611 *G* allele in individuals of Mexican Ancestry from Los Angeles (51%) and Asian individuals (55%) was major when compared with Caucasian and African populations (32%, and 23%, respectively). On the other hand, the frequency of the rs1024611 *A* allele was major in Caucasian and African populations (68%, and 77%, respectively) when compared with Mexican (from Los Angeles), and Asian populations (49% and 45%, respectively). In our study in Mexican individuals, the frequency of the *G* allele was 56% in patients and 55.9% in controls, which is very similar to the frequencies in Mexicans from Los Angeles and Asian populations. Is important to consider that in our study, the concentrations of MCP-1 were determined in frozen samples obtained between 2008 and 2013. The samples were collected at different times in the patients and controls. In patients, usually time after the cardiac event (chronic state). Besides the ethnic differences and the number of individuals included in the studies, this commented fact could be the reason for the differences between our study and those previously reported.

Our study has several strengths. First, it includes a large cohort of Mexican cases and controls well characterized biochemically, clinically, and tomographically, allowing analyses to be adjusted for a considerable number of potentially confounding variables. Second, the control group exclusively includes individuals without tomographic evidence of subclinical atherosclerosis (CAC score = 0). Some limitations of the study should be considered. First, it is not possible to conclude causality due to the cross-sectional nature of the study. Second, it is possible that findings may not apply to the general population because the selection of participants was not random. However, if we consider that the participants do not know their genotypes, the distribution of these would be expected to be similar in a randomly selected sample. Because the Mexican population has a particular genetic background that is different from other ethnic groups, the association observed in the present work must be analyzed in other populations to establish whether they are specific to the Mexican population or shared with other ethnic groups.

## 5. Conclusions

Despite not detecting differences in MCP-1 concentrations between patients with pCAD and control individuals, evaluating concentrations in individuals with the rs1024611 *AA* genotype allowed us to establish that increases of 10 pg/mL of MCP-1 result in a 2.7% increase in the risk of presenting pCAD in these individuals. To our knowledge, this is the first study wherein genetically homogeneous individuals (carriers of the *AA* genotype of the rs1024611 polymorphism) are subject to an increase in the concentrations of a chemokine (MCP-1), which in turn increases the risk of suffering from pCAD, and this is independent of other cardiovascular risk factors such as age, sex, smoking, LDL-C concentrations, presence of small and dense LDL (LDL pattern B), hypertension and elevated levels of high-sensitivity CRP. These results observed in genetically homogeneous subjects could explain, at least in part, the contradictory data that exist in the international literature regarding the association of MCP-1 concentrations and the rs1024611 polymorphism with CAD and pCAD.

## Figures and Tables

**Figure 1 biomedicines-12-01292-f001:**
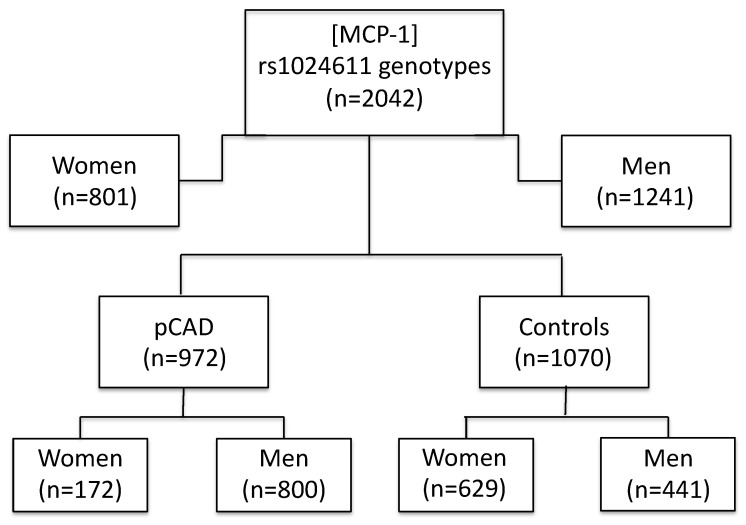
Flowchart representing the distribution of the participants in the study. MCP-1 concentrations and rs1024611 genotypes were determined in 2042 individuals (972 pCAD patients and 1070 healthy controls) belonging to the GEA Mexican Study. GEA: Genetics of Atherosclerotic Disease; pCAD: premature coronary artery disease; and MCP-1: monocyte chemoattractant protein-1.

**Figure 2 biomedicines-12-01292-f002:**
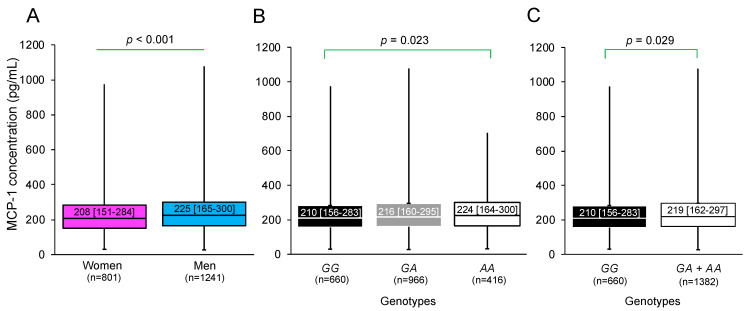
MCP-1 concentrations in the total population stratified by (**A**) sex. (**B**) Genotypes and (**C**) dominant model of the rs1024611 polymorphism of the *MCP-1* gene. The MCP-1 concentrations and rs1024611 genotypes were determined in 2042 individuals. Data are shown as median [interquartile range]. Comparisons between groups were made with the Mann Whitney U or Kruskal–Wallis tests.

**Figure 3 biomedicines-12-01292-f003:**
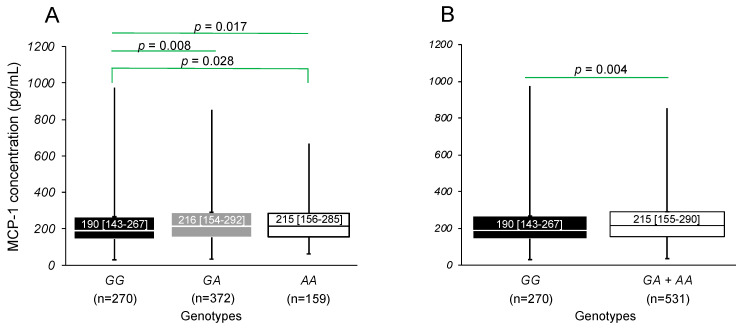
MCP-1 concentrations in women stratified by (**A**) genotypes and (**B**) dominant model of the rs1024611 polymorphism of the *MCP-1* gene. The MCP-1 concentrations and rs1024611 genotypes were determined in 801 women. Data are shown as median [interquartile range]. Comparisons between groups were made with the Mann Whitney U or Kruskal–Wallis tests.

**Figure 4 biomedicines-12-01292-f004:**
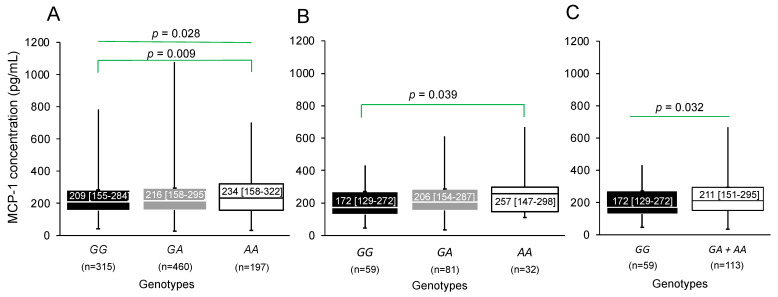
MCP-1 concentrations in patients with pCAD stratified by (**A**) the rs1024611 polymorphism of the *MCP-1* gene, female patients stratified by (**B**) genotypes and (**C**) by the dominant model of the rs1024611 polymorphism of the *MCP-1* gene. The study included 175 women with pCAD, and the MCP-1 concentrations and rs1024611 genotypes were determined in 172 of them. Data are shown as median [interquartile range]. Comparisons between groups were made with the Mann Whitney U or Kruskal–Wallis tests.

**Figure 5 biomedicines-12-01292-f005:**
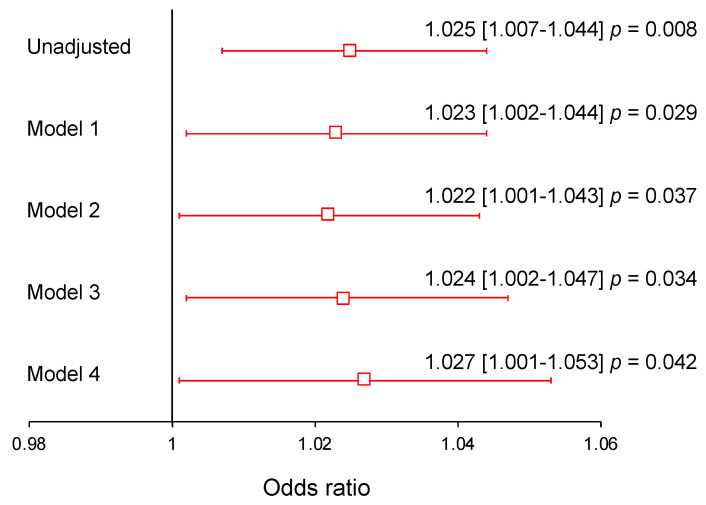
Association between pCAD and MCP-1 concentration in individuals with *AA* genotype of the rs1024611 polymorphism of the *MCP-1* gene. Odds ratios for the association with pCAD are shown for each 10 pg/mL increase in serum MCP-1 concentration. Model 1: adjusted for age and sex. Model 2: adjusted for model 1 + body mass index and smoking. Model 3: adjusted by model 2 + LDL-C concentrations and LDL pattern B. Model 4: adjusted by model 3 + hypertension and high-sensitivity C-reactive protein > 3 mg/L.

**Table 1 biomedicines-12-01292-t001:** Characteristics of the study groups.

	Control	pCAD	*p*
	n = 1070	n = 972	
Demographic and clinical characteristics			
Age (years)	51 (45–57)	53 (49–58)	<0.001
Sex (% de men)	41.2	82.3	<0.001
Body mass index (kg/m^2^)	27.8 (25.4–30.8)	28.3 (26.0–31.1)	0.002
Waist circumference (cm)	94 (86–101)	97 (92–105)	<0.001
Systolic blood pressure (mmHg)	112 (104–123)	115 (106–126)	<0.001
Diastolic blood pressure (mmHg)	71 (65–77)	71 (66–78)	0.159
Biochemical profile			
Total cholesterol (mg/dL)	189 (166–210)	161 (132–193)	<0.001
HDL-Cholesterol (mg/dL)	45(36–55)	37 (32–44)	<0.001
LDL-Cholesterol (mg/dL)	115 (95–134)	91 (69–116)	<0.001
Triglycerides (mg/dL)	145 (107–202)	163 (119–222)	<0.001
LDL size	1.21 (1.08–1.38)	1.12 (0.97–1.32)	<0.001
Glucose (mg/mL)	89 (84–97)	94 (87–117)	<0.001
hsCRP (mg/L)	1.5 (0.80–3.10)	1.15 (0.61–2.54)	<0.001
MCP-1 (pg/mL)	217 (164–288)	217 (157–298)	0.883
Coronary risk factors			
Obesity (%)	29.9	35.1	0.001
Hypertension (%)	18.8	67.9	0.002
Hypercholesterolemia (%)	36.4	20.4	<0.001
Hypoalphalipoproteinemia (%)	51.5	67.5	<0.001
Hypertriglyceridemia (%)	47.3	56.7	<0.001
LDL Pattern B (%)	47.0	60.9	<0.001
Current smoking (%)	22.4	12.2	<0.001
*Genotypes*			
*MCP1* rs1024611 (%) *GG*/*GA*/*AA*	32.2/47.3/20.5	32.4/47.3/20.3	0.993

hsCRP: high sensitivity C-reactive protein. MCP-1: monocyte chemoattractant protein-1. HDL: high-density lipoprotein. LDL: low-density lipoprotein.

## Data Availability

Data will be made available on request.

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
