# Peer review of "MCP-1 rs1024611 Polymorphism, MCP-1 Concentrations, and Premature Coronary Artery Disease: Results of the Genetics of Atherosclerotic Disease (GEA) Mexican Study"

_biomedicines, 2024, doi:10.3390/biomedicines12061292_

Round 1
Reviewer 1 Report
Comments and Suggestions for Authors
The presented paper describes a specific medical experiment. I appreciate only the statistical side of the job. The statistical part of the work is written in an orderly manner following well-known algorithms. Statistically speaking, there is nothing new at work. Old standard research methods are used, as in many similar medical articles. Meanwhile, I hope the biomedical side will be properly evaluated by specialists in that field. From my side, I have no complaints about the work.
Author Response
Thank you for your comments.Reviewer 2 Report
Comments and Suggestions for Authors
The proposed manuscript is devoted to the results of a research aimed to define whether monocyte chemoattractant protein-1 concentrations are associated with the presence of premature coronary artery disease and to establish whether variations in the rs1024611 polymorphism increase monocyte chemoattractant protein-1 concentrations in specific groups of patients.
Preliminaries to the research area are provided. Information related to the role of inflammation in the progression of atherosclerosis and in particular the role of monocyte chemoattractant protein-1/CCL2 in the formation, progression, and destabilization of atherosclerotic plaques is reviewed.
The methodology of the proposed study is described in detail, in particular the subjects of the study, the parameters and life-style characteristics assessed in the participants, details of the in silico and statistical analyses.
The results of the study are presented and illustrated by many tables and figures. A detailed discussion of the results is provided and reasonable conclusions are formulated.
The presentation of the main results is clear and comprehensive. From a formal point of view, all the contents seems to be correct. The results are valuable and worthy of being published taking into account their possible applications in clinical practice and health care.
Minor revisions are suggested to improve the quality of the exposition:
p. 1, line 1: The title of the manuscript seems too long and too descriptive. I recommend some reasonable simplification to be done.
p. 1, line 42: The open parenthesis before the left square bracket in “([1]” should be removed.
p. 3, line 95: The period before “with pCAD (p>0.05)” should be removed.
p. 8, line 253: I suggest to move the comma after the source number: “was observed by Hwang et al [39], who” instead of “was observed by Hwang et al, [39] who”.
Author Response
Answer: The title has been modified.
MCP-1 rs1024611 polymorphism, MCP-1 concentrations, and premature coronary artery disease. Results of the GEA Mexican study.
p. 1, line 42: The open parenthesis before the left square bracket in “([1]” should be removed.
Answer: The open parenthesis has been removed.
p. 3, line 95: The period before “with pCAD (p>0.05)” should be removed.
Answer: The period before “with pCAD (p>0.05)” has been removed.
p. 8, line 253: I suggest to move the comma after the source number: “was observed by Hwang et al [39], who” instead of “was observed by Hwang et al, [39] who”.
Answer: The comma after the source number: “was observed by Hwang et al [39], who” has been moved.
Reviewer 3 Report
Comments and Suggestions for Authors
- the authors should re-assess the text. The iThenticate index is pretty high. Even though most sentences that are flagged are pretty generic, they suggest a minor form of mosaic plagiarism, and should be avoided.
- line 70: the objective should be rewritten - maybe which MCP-1 concentrations are associated with pCAD...
-chapter 2.1 - were the two groups matched or not? If yes, based on which parameters?
- why was used the Mann-Whitney U test? Was the assumption of normality not present? The number of cases is pretty high. What test was used to assess the normality of the distribution? What statistical significance level was used?
- from table 1 it is obvious that an improper selection of the subject was able to influence the results of the study. When comparing a control group with only 41.2% males without pCAD with a group containing 82.3% subjects with pCAD, the MCP-1 values were basically equal. To properly evaluate the MCP-1 values, and differences, the authors should have matched the groups based on gender (and also potentially age). Alternatively, they should have employed more robust statistical methods (e.g. multivariate analyses, logistic regression, to control for the unmatched groups)
- the discussion should be reevaluated after correcting the methodological errors.
Comments on the Quality of English Language
- moderate corrections should be made
Author Response
- the authors should reassess the text. The iThenticate index is pretty high. Even though most sentences that are flagged are pretty generic, they suggest a minor form of mosaic plagiarism, and should be avoided.
Answer: The manuscript has been revised to detect plagiarism and the software used reported a 6.8% similarity score.
- line 70: the objective should be rewritten - maybe which MCP-1 concentrations are associated with pCAD...
Answer: The objective has been rewritten: Thus, the objective of the present study is to define if MCP-1 concentrations and MCP-1 rs1024611 genotypes are associated with the presence of pCAD in Mexican mestizo individuals from the GEA (Genetics of Atherosclerotic Disease) Study.
-chapter 2.1 - were the two groups matched or not? If yes, based on which parameters?
Answer: The groups were not matched. All individuals belong to the GEA Mexican Study cohort. The characteristics of the two groups are described in the material and methods section.
- why was used the Mann-Whitney U test? Was the assumption of normality not present? The number of cases is pretty high. What test was used to assess the normality of the distribution? What statistical significance level was used?
Answer: Before performing the analyses, we test the normality distribution of the variables with the Kolgomorov-Smirnov test. When the p-value was <0.05, the variable distribution was considered not normal. None of the variables included in the present manuscript have a normal distribution. Therefore, the comparison between the control and pCAD groups was done using the Mann-Whitney U test.
We considered significant differences when the value was <0.05.
- from table 1 it is obvious that an improper selection of the subject was able to influence the results of the study. When comparing a control group with only 41.2% males without pCAD with a group containing 82.3% subjects with pCAD, the MCP-1 values were basically equal. To properly evaluate the MCP-1 values, and differences, the authors should have matched the groups based on gender (and also potentially age). Alternatively, they should have employed more robust statistical methods (e.g. multivariate analyses, logistic regression, to control for the unmatched groups)
Answer: As we previously mentioned, the participants included in the present study belong to the GEA Mexican Study. We did not make a selection of the participants, we included the subjects in which, it was possible to determine the MCP-1 concentrations and the MCP-1 genotypes.
Considering that pCAD is more frequent in men than in women, we used the logistic regression analysis to properly evaluate the participation of MCP-1 concentration in pCAD, as suggested by the reviewer, adjusting for potentially confounding variables. As described in the Figure 5 footnote, the models were adjusted by age, sex, body mass index, LDL-cholesterol, LDL pattern B, high-sensitivity CRP, hypertension, and current smoking.
- the discussion should be reevaluated after correcting the methodological errors.
Answer: As was commented in the previous point, we used a logistic regression analysis adjusted by confounding variables. So, we consider that the results reported are correct and the discussion according to those results.
Comments on the Quality of English Language
- moderate corrections should be made
Answer: The text has been revised for the English language.
Reviewer 4 Report
Comments and Suggestions for Authors
The article is very good, the main advantage being the large number of patients, 972 patients with pCAD and 1070 healthy controls. The results of this study showed that for every ten pg/mL increase in MCP-1 concentration, the risk of presenting pCAD increases by 2.7% in AA genotype individuals. Individuals with the MCP-1 rs1024611 AA genotype present an increase in MCP-1 concentration. In those individuals, increased MCP-1 concentrations increase the risk of presenting pCAD. However, there are some small points that could be improved:
1 - The figures need to be improved, the legends are too small and some numbers cannot be read.
2 - Table 1 needs to be improved in format and there are several acronyms that are not defined.
Author Response
1 - The figures need to be improved, the legends are too small and some numbers cannot be read.
Answer: The figures have been improved. The size of the numbers has been increased
2 - Table 1 needs to be improved in format and there are several acronyms that are not defined.
Answer: The table has been improved in format and the acronyms have been defined.
Round 2
Reviewer 3 Report
Comments and Suggestions for Authors
The authors responded to some comments only in the response to reviewers, without making the corresponding changes in the manuscript. For example, the explanation regarding normality testing using the KS test was not included in the materials and methods.
Comments on the Quality of English Language
English is fine, there are only a few minor issues
Author Response
Reviewer 3
The authors responded to some comments only in the response to reviewers, without making the corresponding changes in the manuscript. For example, the explanation regarding normality testing using the KS test was not included in the materials and methods.
Answer: We agree with the reviewer and to clarify this point, the phrase “Before performing the analysis, we test the normality distribution of the variables with the Kolmogórov-Smirnov test. When the p-value was <0.05, the variable distribution was considered not normal.” has been added in the material and methods section.
and the paragraph “We considered significant differences when the p-value was <0.05.” has also been added.
In the footnotes of the figures, is indicated the test used in each case.
Comments on the Quality of English Language
English is fine, there are only a few minor issues
Answer: A few minor issues have been corrected.
Round 3
Reviewer 3 Report
Comments and Suggestions for Authors
I feel we are circling here. Now, the authors have said that they used the KS test to test for normality. But what were the results? Should we believe them? Taking into account the number of cases, and the central limit theorem, I feel the normality assumption should have been fulfilled. This is a scientific article, and the results should be proved by data.
Comments on the Quality of English LanguageEnglish is fine.
Author Response
Response to reviewer 3
We agree with the reviewer and in order to clarify this point, the phrase “Before performing the descriptive analysis, we test the distribution of the continuous variables with the Kolmogorov-Smirnov test. When the p-value was <0.05, the variable distribution was considered asymmetrical” has been added in the material and methods section.
We are providing the table with the full Kolmogorov-Smirnov test
|
Continuous variable |
Statistic |
|
p_Value |
Symmetry |
Decision |
|
Age (years) |
0.047 |
|
<0.001 |
Asymmetrical |
Mean and IQR |
|
Body mass index (kg/m2) |
0.035 |
|
<0.001 |
Asymmetrical |
Mean and IQR |
|
Waist circumference (cm) |
0.024 |
|
<0.001 |
Asymmetrical |
Mean and IQR |
|
Systolic blood pressure (mmHg) |
0.079 |
|
<0.001 |
Asymmetrical |
Mean and IQR |
|
Diastolic blood pressure (mmHg) |
0.052 |
|
<0.001 |
Asymmetrical |
Mean and IQR |
|
Total cholesterol (mg/dL) |
0.041 |
|
<0.001 |
Asymmetrical |
Mean and IQR |
|
HDL-Cholesterol (mg/dL) |
0.089 |
|
<0.001 |
Asymmetrical |
Mean and IQR |
|
LDL-Cholesterol (mg/dL) |
0.041 |
|
<0.001 |
Asymmetrical |
Mean and IQR |
|
Triglycerides (mg/dL) |
0.122 |
|
<0.001 |
Asymmetrical |
Mean and IQR |
|
LDL size |
0.074 |
|
<0.001 |
Asymmetrical |
Mean and IQR |
|
Glucose (mg/mL) |
0.262 |
|
<0.001 |
Asymmetrical |
Mean and IQR |
|
CRP high sensitivity (mg/L) |
0.314 |
|
<0.001 |
Asymmetrical |
Mean and IQR |
|
MCP-1 (pg/mL) |
0.088 |
|
<0.001 |
Asymmetrical |
Mean and IQR |
Round 4
Reviewer 3 Report
Comments and Suggestions for Authors
The authors corrected the manuscript according to the comments of the reviewers.
Comments on the Quality of English LanguageEnglish is fine